# Exploring concurrent validity of the CLN2 Clinical Rating Scale: Comparison to PedsQL using cerliponase alfa clinical trial data

Nicola Specchio[1]*, Paul Gissen[2], Emily de los Reyes[3], Andrew Olaye[4¤a], Charlotte Camp[4], Tristan Curteis[5], Annabel Griffiths[6], Thomas Butt[4], Jessica Cohen-Pfeffer[7], Peter Slasor[7], Zlatko Sisic[4¤b], Mohit Jain[4], Angela Schulz[8]

1 Department of Neuroscience, Bambino Gesù Children's Hospital, IRCCS, Rome, Italy, 2 NIHR Biomedical Research Centre, UCL Great Ormond Street Institute of Child Health, London, United Kingdom, 3 Department of Pediatric Neurology, Nationwide Children's Hospital, Columbus, Ohio, United States of America, 4 BioMarin Europe Ltd, London, United Kingdom, 5 Costello Medical, Manchester, United Kingdom, 6 Costello Medical, Cambridge, United Kingdom, 7 BioMarin Pharmaceutical Inc., Novato, California, United States of America, 8 Department of Pediatrics, Children's Hospital, University Medical Center Hamburg-Eppendorf, Hamburg, Germany

¤a Current address: UK Orchard Therapeutics Limited, London, United Kingdom
¤b Current address: Batten Disease Family Association, London, United Kingdom
* nicola.specchio@gmail.com

**Data Availability Statement:** Results of the open-label, phase 1/2 study (Study 190–201; NCT01907087) and extension study (Study 190–

## Abstract

### Background

The CLN2 Clinical Rating Scale evaluates disease progression in CLN2 disease, an ultra-rare, neurodegenerative disorder with late infantile onset. To validate the Clinical Rating Scale, a comparison with the Pediatric Quality of Life Inventory (PedsQL) was conducted utilising clinical trial data investigating cerliponase alfa use in CLN2 disease.

### Methods

Linear regression and mixed effects models were used to investigate the relationship between the Clinical Rating Scale and PedsQL using open-label, single-arm, phase 1/2 (NCT01907087) and ongoing extension study (NCT02485899) data of 23 children with CLN2 disease treated with cerliponase alfa for ≥96 weeks.

### Results

Correlations between the four Clinical Rating Scale domains were low. Linear mixed effects analyses showed significant correlation between PedsQL and Clinical Rating Scale (Total score or motor-language [ML] score adjusted p-values <0.05), driven by the relationship with the PedsQL Physical domain. A statistically significant relationship was identified between the Clinical Rating Scale motor domain and PedsQL (Total score: adjusted p-value = 0.048, parameter estimate [PE] = 8.10; Physical domain score: adjusted p-value = 0.012; PE = 13.79).

202; NCT02485899) of cerliponase alfa are available at: https://clinicaltrials.gov/study/NCT01907087 and https://clinicaltrials.gov/study/NCT02485899. To protect patient privacy and due to ethical restrictions, BioMarin will provide secure access to anonymized patient-level data that underlie the results reported in this article to qualified researchers in response to scientifically valid research proposals. We confirm that interested researchers can request this data at the following link: www.BioMarin.com/patients/publication-data-request/.

**Funding:** BioMarin Pharmaceutical Inc funded this study, and were involved in the design of the study, analysis and interpretation of data, as well as in preparation of this manuscript and the decision to submit this article for publication. Medical writing and editorial assistance in preparing this manuscript for publication was provided by Costello Medical, UK, and funded by BioMarin Pharmaceutical Inc.

**Competing interests:** PG, NS: No competing interests to declare; AS: Received compensation from BioMarin Europe as per clinical trial agreement; EdIR: Research grants from Amicus Therapeutics, BioMarin Pharmaceutical Inc., Giving Back Fund; Consultancy: BioMarin Pharmaceuticals Inc.; AO, CC, TB, and MJ: Employees and shareholders of BioMarin Europe; ZS: Employee of BioMarin Europe; TC and AG: Employees of Costello Medical; JCP and PS: Employees and shareholders of BioMarin Pharmaceutical Inc. This does not alter our adherence to PLOS ONE policies on sharing data and materials.

## Conclusions

Each domain of the Clinical Rating Scale provides unique information on disease state. Validity of the scale is supported by its relationship with the PedsQL. Among the four domains of the Clinical Rating Scale, motor has the highest correlation to PedsQL, suggesting motor function as a driver of patients' quality of life. The lack of association between the remaining domains of the Clinical Rating Scale and PedsQL suggests that additional disease-specific measures may be needed to fully capture the quality of life impact of CLN2 disease.

## Trial registration

NCT01907087, NCT02485899.

## Introduction

Neuronal ceroid lipofuscinosis type 2 (CLN2) disease is an ultra-rare, autosomal recessive disorder typically presenting in children aged between 2 and 4 years of age [1]. CLN2 disease has an estimated prevalence of 0.75 per million population and incidence of 0.5 per 100,000 live births based on literature reports [2]. CLN2 disease is caused by mutations in the *CLN2* gene leading to deficiency of the lysosomal proteinase known as tripeptidyl peptidase 1 (TPP1) [3]. Deficient TPP1 activity leads to intralysosomal accumulation of autofluorescent storage material and is associated with neuronal and retinal cell loss [4]. The classic phenotype of CLN2 disease generally manifests with new-onset seizures and/or ataxia, typically in combination with a history of early language delay [5]. This is followed by rapid disease progression including progressive language loss, movement disorders, dementia, seizures, myoclonus and visual deterioration [6, 7]. Ultimately, patients with CLN2 disease experience reduced life expectancy, with a median age at death reported as 10.0 years [6, 7]. The natural course of the condition over time is described in Fig 1.

The Hamburg scale is a CLN2 disease-specific rating scale, which was developed to quantify the loss of function that occurs as CLN2 disease progresses [6]. The four-item instrument assesses motor function (walking ability), visual function, language and seizures, with each item scored 0–3 to give a total combined score between 0 and 12. The four-domain CLN2 Clinical Rating Scale, hereafter referred to as the Clinical Rating Scale, is an adapted form of the Hamburg scale designed to allow consistent ratings in multinational, multisite, clinical efficacy studies [6, 8, 9]. The Clinical Rating Scale is comprised of the visual function and seizure items of the Hamburg scale and adapted versions of the motor function and language items (Table 1).

For each domain of the Hamburg and Clinical Rating Scales, a score from 0–3 represents age-appropriate best function (3) to essentially no function (0). The summation of the domain scores leads to a four-domain Total score between 0 and 12 [9]. The Clinical Rating Scale was devised to enable prospective acquisition of clinical trial data that could be compared to existing historical data [8]. Changes to the items included the description of a motor score of 2 from "frequent falls, obvious clumsiness" in the Hamburg scale to the more specific definition of "abnormal gait; independent ≥10 steps; frequent falls, obvious clumsiness" in the Clinical Rating Scale. The Motor-Language (ML) scale is the sum of the Motor and Language domains (score 0–6) and a primary measure of CLN2 disease progression. The ML score has

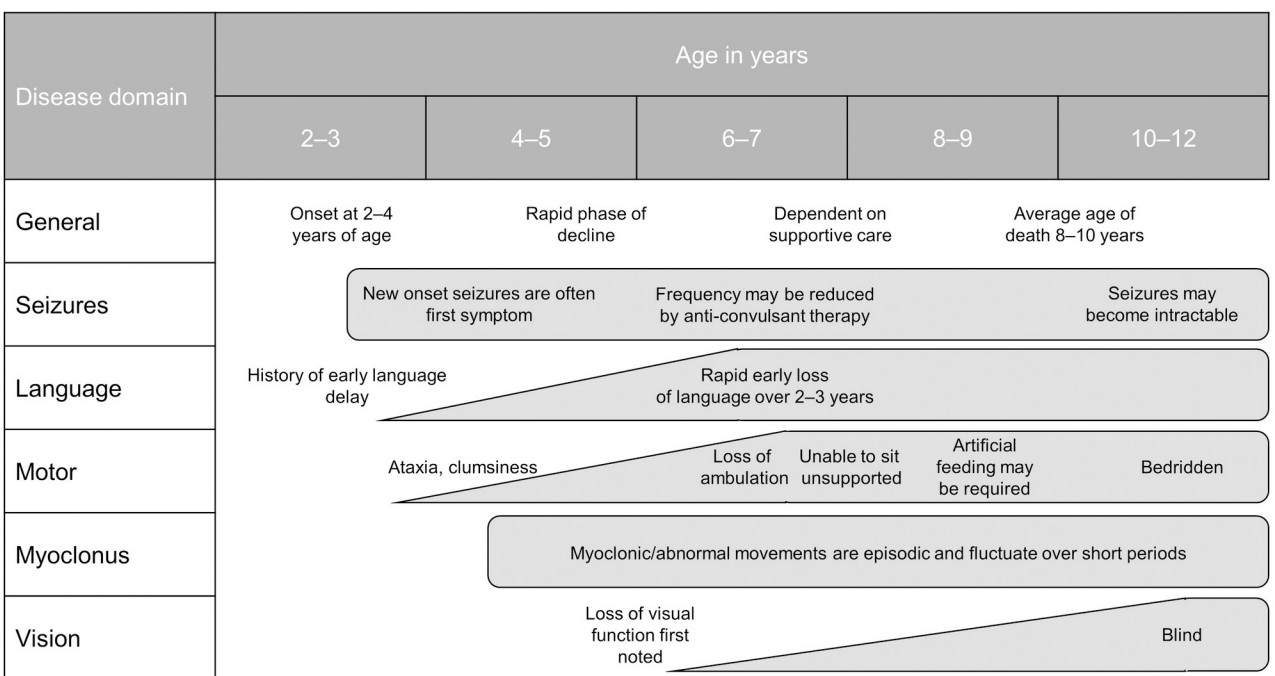

**Fig 1. Natural course of CLN2 disease.** CLN2: Neuronal ceroid lipofuscinosis type 2. Sources: Steinfeld R, et al. 2002 [6] and Nickel M, et al. 2018 [7] image.

demonstrated adequate similarity to the equivalent items of the Hamburg scale, based on their application in treated and untreated CLN2 populations [8].

The Clinical Rating Scale was the primary outcome measure in the open-label, phase 1/2 study (Study 190–201; NCT01907087) and ongoing extension study (Study 190–202;

**Table 1. Comparison between the Hamburg Scale and Clinical Rating Scale.**

| Domain/Item | Score | Hamburg Scale [6, 8] | Clinical Rating Scale [8, 9] |
|---|---|---|---|
| Motor | 3 | Walks normally | Grossly normal gait |
| | 2 | Frequent falls, obvious clumsiness | Abnormal gait; independent ≥10 steps; Frequent falls, obvious clumsiness |
| | 1 | No unaided walking or crawling only | |
| | 0 | Immobile, mostly bedridden | |
| Language | 3 | Normal (individual maximum) | Grossly normal |
| | 2 | Has become recognisably abnormal | Has become recognisably abnormal (worse than the individual maximum) |
| | 1 | Hardly understandable | |
| | 0 | Unintelligible or no language | |
| Visual | 3 | Recognises desirable object, grabs at it | |
| | 2 | Grabbing for objects uncoordinated | |
| | 1 | Reacts to light | |
| | 0 | No reaction to visual stimuli | |
| Seizures | 3 | No seizure in 3 months | |
| | 2 | 1–2 seizures in 3 months | |
| | 1 | 1 seizure per month | |
| | 0 | >1 seizure per month | |

NCT02485899) of the enzyme replacement therapy cerliponase alfa, a recombinant form of human TPP1, which is administered by an intracerebroventricular (ICV) infusion [9–11]. These studies have demonstrated a clinically significant difference between cerliponase alfa treatment and historical controls; treatment with cerliponase alfa was shown to result in a slower rate of decline of motor and language function in CLN2 disease patients [9]. Quality of life (QoL), as measured by tools such as the PedsQL (Pediatric Quality of Life Inventory), was also included as an exploratory efficacy outcome of the treatment studies. Cerliponase alfa was approved for use in paediatric patients with CLN2 disease by both the European Medicines Agency (EMA) and the U.S. Food and Drug Administration (FDA) in 2017 [12, 13]. Prior to cerliponase alfa, standard of care had been symptomatic and palliative.

Disease characteristics thought to impact patient QoL may be captured in patient-reported outcome (PRO) measures. For example, the PedsQL is an established, generic measure for evaluating QoL in children and adolescents aged between 2–18 years and is available in versions suitable for completion by the child (self-report) or on their behalf, for example by a parent (proxy-report) [9, 14–18]. The PedsQL 4.0 Generic Core Scales, hereafter referred to as PedsQL, comprises of four domains: Physical Functioning, Emotional Functioning, Social Functioning, and School Functioning, which are subsequently referred to as the Physical, Emotional, Social and School domains [15, 17]. Each domain is scored separately, and a Total score across the multiple domains is also calculated. Possible scores, both for individual domains and the Total score, range from 0 to 100, where 0 is the least favourable score and 100 is the most favourable score. The PedsQL offers the advantages of being brief (typically taking less than 5 minutes to complete) and having an established minimal clinically important difference (MCID) of 4.5 [15, 16, 19]. The PedsQL has been used to assess QoL in patients with CLN2 disease but has not been specifically validated in this patient population.

A previous study by Wyrwich et al. examined construct validity of the 2-domain ML scale by using correlational analyses to compare motor domain scores to the PedsQL (Physical and Total scores), and ML scores to PedsQL Total scores, using baseline data from the cerliponase alfa treatment studies [8]. Some evidence of moderate to strong baseline correlations were identified. This study builds on the findings of Wyrwich et al. in providing further validation of the Clinical Rating Scale, through utilising all available trial data to investigate the following: correlations between all four domains of the scale; the relationship of the scale to all domains of the PedsQL; and the independent contributions of each domain to any identified relationship. The application of mixed effects analyses was also used to account for within-patient correlation. These analyses enable an understanding of the most patient-relevant aspects of CLN2 disease and provide insights into the usefulness of the PedsQL in the CLN2 patient population.

## Methods

### Data used for analyses

Data from the open label, single-arm, phase 1/2 study (Study 190–201; NCT01907087) and ongoing extension study (Study 190–202; NCT02485899) of 23 CLN2 disease patients treated with cerliponase alfa for a minimum of 96 weeks were used in these analyses (24 patients were enrolled, however, at the parents' request, one patient withdrew after the receipt of one dose of the study drug owing to an unwillingness to continue with study visits and procedures, and was therefore excluded from this analysis; Fig 2); hereafter referred to as Study 201/202 data [9–11].

In Study 190–201, the first safety cohort (n = 10) underwent dose escalation for up to 23 weeks to establish an acceptable side-effect profile [9]. Subsequently, all patients initiated stable dosing (SD), which consisted of a fixed dose of 300 mg every 14 days given via an ICV infusion

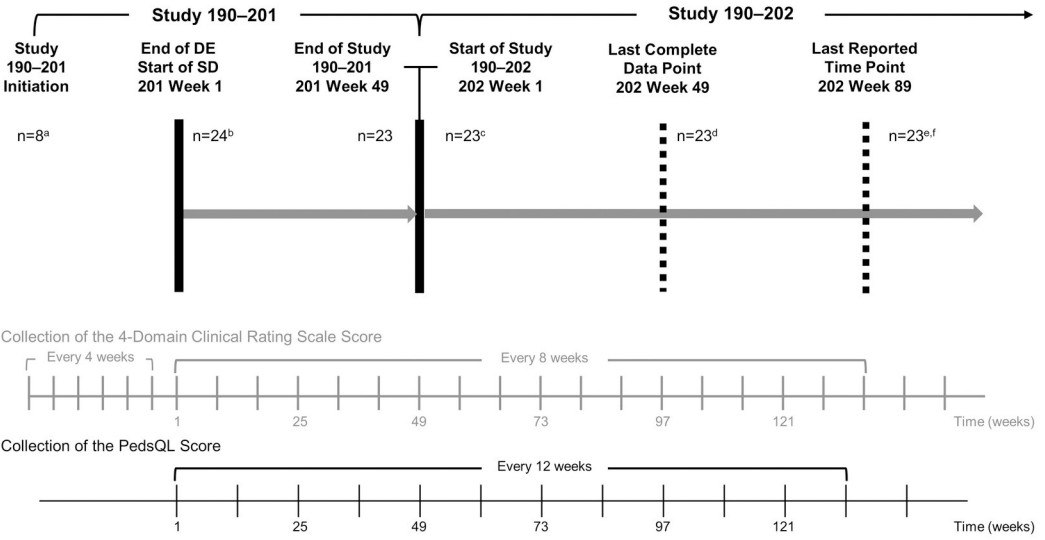

**Fig 2. Study 190-201/202 summary diagram.** [a]10 patients had a DE period ranging from 5–23 weeks. [b]14 patients enrolled directly to the 300 mg SD for 48 weeks of treatment/assessment. At the parents' request, one patient withdrew after the receipt of one dose of the study drug owing to an unwillingness to continue with study visits and procedures; the other 23 patients completed the dosing and follow-up for Study 190–201 [9]. [c]23 patients continued onto Study 190–202 receiving their first dose immediately after the Week 49 assessment in Study 190–201; [d]Data are available for all patients up to Week 49 of Study 190–202; [e]Patients in Study 109–202 continue treatment for up to 239 weeks. DE: dose escalation; SD: stable dose.

for 4–5 hours. Patients who joined the study later went directly into SD treatment. Completers of Study 190–201 were then enrolled in Study 190–202 and continued SD treatment [9]. Written informed consent from a parent or legal guardian of each patient was obtained, and assent was obtained from the patient, if appropriate. The studies were performed in accordance with the provisions of the Declaration of Helsinki. Approval was obtained from the institutional review board at each participating centre [9].

Data were available for all patients up to 96 weeks of treatment at 300 mg cerliponase alfa dosing for the clinician-reported Clinical Rating Scale and the proxy-reported PedsQL 4.0 Generic Core Scales (Parent Report for Toddlers) [PedsQL™, Copyright © 1998 JW Varni, Ph. D. All rights reserved]. Clinical Rating Scale data were collected every 8 weeks and PedsQL data every 12 weeks, leading to concurrent data for both measures being available every 24 weeks (Fig 2).

## Statistical analyses

**Assessing the relationships between the domains of the Clinical Rating Scale.** The relationships between the domains of the Clinical Rating Scale were first visually inspected as scatter plots and Spearman correlation coefficients (as well as bootstrap p-values) were calculated using data from all timepoints. The relationships were subsequently assessed using simple and multiple linear regression. Variance inflation factors (VIFs) and condition indices were calculated, which are standard measures of (multi)collinearity [20]. A VIF describes how much the variance of the regression coefficients can be explained by the correlation between two or more variables, compared to a model where there is no correlation between these variables. A condition index provides additional information to the VIF, indicating how possible it is to explain one or more variables as a weighted, linear combination of other variables. When the

condition index is high, it is important to assess the relative contribution of the variable of interest to the variance of the other variables. The following VIF thresholds were assumed in these analyses: VIF $\geq$ 10 indicates high multicollinearity present between variables; $5 \leq$ VIF $<10$ indicates moderate multicollinearity present between variables. A condition index $\geq$ 30 indicates high multicollinearity between some or all variables within the model, while common conventions suggest a condition index from 10–30 indicates low/medium multicollinearity [20].

**Investigating the relationship between the Clinical Rating Scale and PedsQL.** Linear mixed effects analyses, which accounted for within-patient correlation, were conducted to assess the relationships between the PedsQL (domain and Total score) and the Clinical Rating Scale (ML and Total score). The fixed effects consisted of the Clinical Rating Scale ML or Total score. In all linear mixed effects models, a selection of random effect structures was considered, including only random intercepts, only random slopes, and both random intercepts and slopes. The Akaike information criterion was used to select the final model. P-values were obtained from Kenward-Roger adjusted F-tests and were Holm-Bonferroni corrected to adjust for multiple comparisons. Confidence intervals (CIs) were Wald-type approximations.

**Exploring the contributions of the domains of the Clinical Rating Scale.** Multivariate linear mixed effects analyses were conducted to understand the individual contributions of the domains of the Clinical Rating Scale to the relationship with the PedsQL. The individual domains were used as independent variables in the same regression models; the motor, language, vision and seizure domains were entered as fixed effects in the multivariate analyses and a choice of random slopes considered. In addition, the motor and language domains alone were tested within the same regression model, in which the fixed effects consisted of these domains only.

Statistical analyses were conducted using R version 3.5.3 [21].

## Results

### Relationships between the domains of the Clinical Rating Scale

As a first analysis, scatter plots presenting domain scores of the Clinical Rating Scale from all timepoints of Study 201/202 were visually inspected. These scatter plots indicated some evidence of correlation between the motor, language and vision domains (Fig 3 and Table 2). Results of the multicollinearity analyses revealed VIF estimates which were considerably below the threshold for moderate multicollinearity ($5 \leq$ VIF $<10$), demonstrating low correlation among these domains (all VIFs $<5$) (Table 3).

Pairwise domain comparisons were assessed to determine whether collinearity existed between any two domains. In all cases, VIFs were below the threshold required to provide evidence of collinearity (Table 4). To further quantify the relationship between the domains, condition indices were calculated. The largest condition index from all comparisons was below the assumed threshold ($\geq$ 30) required to provide evidence of (multi)collinearity (Table 5).

### Relationship between the Clinical Rating Scale and PedsQL

Linear mixed effects models demonstrated that there was evidence of a statistically significant correlation between the PedsQL and the Total score (adjusted p-value = 0.004; PE = 2.56) or the ML score (adjusted p-value = 0.004; PE = 4.90) of the Clinical Rating Scale (Table 6). Furthermore, these relationships seem to be driven by the relationship with the Physical domain of the PedsQL, which was the only domain to show a statistically significant relationship with the Total score (adjusted p-value = 0.010) and the ML score (adjusted p-value$<$0.001) of the Clinical Rating Scale (Table 6). In addition, there was evidence to suggest that a 1-point change

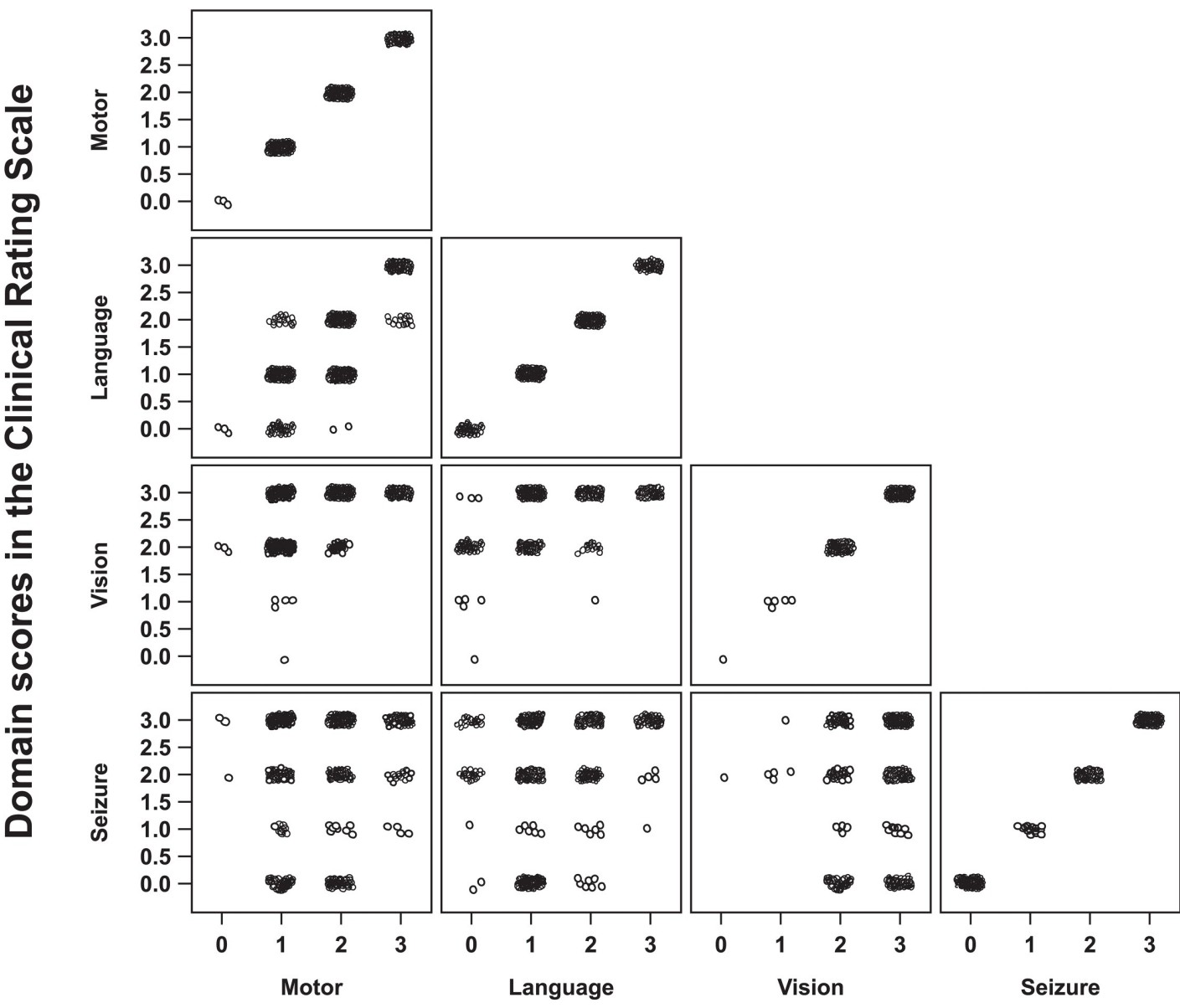

**Fig 3. Scatter plots of domain scores of the Clinical Rating Scale from Study 201/202.**

in the ML score of the Clinical Rating Scale has a patient-relevant impact on QoL as the parameter estimates (PEs) for both PedsQL Total score (PE = 4.90) and Physical domain score (PE = 9.24) were larger than the 4.5-point MCID established for the PedsQL (Table 6) [15].

### Contributions of the domains of the Clinical Rating Scale

In further support of these findings, linear mixed effects models analysing the relationship between each domain of the Clinical Rating Scale and the PedsQL confirmed a statistically

**Table 2. Spearman correlation coefficients of domain scores of the Clinical Rating Scale from Study 201/202.**

| | Spearman correlation coefficient (p-value*) | | | |
| --- | --- | --- | --- | --- |
| | **Motor** | **Language** | **Vision** | **Seizure** |
| **Motor** | 1.00 | 0.63 (<0.001) | 0.51 (<0.001) | 0.07 (0.143) |
| **Language** | 0.63 (<0.001) | 1.00 | 0.42 (<0.001) | 0.19 (<0.001) |
| **Vision** | 0.51 (<0.001) | 0.42 (<0.001) | 1.00 | 0.07 (0.190) |
| **Seizure** | 0.07 (0.143) | 0.19 (<0.001) | 0.07 (0.190) | 1.00 |

*P-values were calculated using the bootstrap approach.

**Table 3. Variance inflation factors for the domains of the Clinical Rating Scale—Study 201/202.**

| Domain of interest | VIF |
| --- | --- |
| Motor | 2.13 |
| Language | 2.05 |
| Vision | 1.33 |
| Seizure | 1.03 |

$5 \leq$ VIF $<10$ is indicative of moderate multicollinearity; results that met this criterion are highlighted in **bold**. VIF: variance inflation factor.

**Table 4. Variance inflation factors for all pairwise domain comparisons of the Clinical Rating Scale—Study 201/202.**

| Pairwise comparison | VIF |
| --- | --- |
| Motor and language | 1.96 |
| Motor and vision | 1.30 |
| Motor and seizure | 1.01 |
| Language and vision | 1.22 |
| Language and seizure | 1.03 |
| Vision and seizure | 1.00 |

$5 \leq$ VIF $<10$ is indicative of moderate multicollinearity; results that met this criterion are highlighted in **bold**. VIF: variance inflation factor.

**Table 5. Condition indices for the domains of the Clinical Rating Scale—Study 201/202.**

| Domains of interest | Largest condition index[a] |
| --- | --- |
| Motor, language, vision and seizure | 2.69[b] |
| Motor and language | 7.57 |
| Motor and vision | 1.69[b] |
| Motor and seizure | 6.45 |
| Language and vision | 1.58[b] |
| Language and seizure | 5.16 |
| Vision and seizure | 1.05[b] |

[a]No values were above the condition index threshold of 30, indicating no significant multicollinearity between some or all variables within the models
[b]Values presented are condition indices for models without the intercept.

**Table 6. Results of the linear mixed effects models comparing the PedsQL to the total and ML scores of the Clinical Rating Scale.**

| Dependent variable (PedsQL) | Independent variable (Clinical Rating Scale) | Marginal $R^2$ | Parameter estimate | 95% CI lower limit | 95% CI upper limit | Adjusted p-value* |
|---|---|---|---|---|---|---|
| **Total score** | Total score[a] | 0.15 | 2.56 | 1.25 | 3.87 | **0.004** |
| | ML score[a] | 0.21 | 4.90 | 2.47 | 7.34 | **0.004** |
| **Physical score** | Total score[b, c] | 0.13 | 3.68 | 1.69 | 5.67 | **0.010** |
| | ML score[b, c] | 0.32 | 9.24 | 6.17 | 12.31 | **<0.001** |
| **Emotional score** | Total score[a] | 0.02 | 1.18 | -0.49 | 2.85 | 0.907 |
| | ML score[b] | 0.03 | 2.13 | -0.60 | 4.86 | 0.863 |
| **Social score** | Total score[b] | 0.05 | 1.86 | 0.23 | 3.48 | 0.294 |
| | ML score[a] | 0.02 | 1.77 | -1.43 | 4.96 | ≈1 |
| **School score** | Total score[c] | 0.01 | 0.84 | -0.89 | 2.56 | ≈1 |
| | ML score[b, c] | 0.00 | -0.70 | -3.25 | 1.85 | ≈1 |

*P-values are Holm-Bonferroni corrected; obtained from Kenward-Roger adjusted F-tests; considered statistically significant and highlighted in **bold** if less than 0.05.

CI, Confidence intervals are Wald type approximations.

[a]Random slopes for this independent variable included in model.

[b]Random intercepts included in model.

[c]Random slopes for time variable included in model.

significant relationship between the motor domain and both the PedsQL Total score (adjusted p-value = 0.048; PE = 8.10) and Physical domain score (adjusted p-value = 0.012; PE = 13.79) (Table 7). The PEs were again larger than the 4.5-point MCID for PedsQL suggesting a 1-point change in the motor domain alone has a patient-relevant impact on QoL. The language, vision and seizure domains were not found to have a statistically significant relationship with the Total score or domain scores of the PedsQL (Table 7).

To further examine the individual relationships of the motor and language domains of the Clinical Rating Scale with the PedsQL, multivariate linear mixed effects analyses were also conducted. Vision and seizure domains were removed from the model in order to increase the likelihood of being able to detect a relationship; the motor and language domains were used as fixed effects within the same regression model. These results confirmed that the motor domain was a statistically significant addition to the PedsQL Total score model (adjusted p-value = 0.023), whereas no significant contribution was observed from the language domain (adjusted p-value = 0.274) (Table 8).

## Discussion

In this study, a series of statistical analyses were used to investigate the relationship between the domains of the Clinical Rating Scale and to further validate this scale by examining its relationship with the established PedsQL measure. The estimated correlations between the four domains of the Clinical Rating Scale from the Study 201/202 data were low, suggesting that each domain contributes novel information. This may be explained by the respective timing and progression of the different aspects of CLN2 disease. For example, it has been shown that for CLN2 disease patients with the classic phenotype the language domain score deteriorates faster, and to a greater degree, than the motor domain, while vision loss is observed at later stages only [7, 9]. As such, it is understandable that the levels of correlation between the domain scores were low.

Linear mixed effects analysis provided evidence of concurrent validity between the Clinical Rating Scale and PedsQL, with results suggesting that the Physical domain of the PedsQL

**Table 7. Results of the linear mixed effects models comparing the PedsQL to the domains of the Clinical Rating Scale.**

| Dependent variable (PedsQL) | Independent variable (Clinical Rating Scale) | Marginal $R^2$ | Parameter estimate | 95% CI lower limit | 95% CI upper limit | Adjusted p-value* |
|---|---|---|---|---|---|---|
| **Total score** | Motor | 0.24 | 8.10 | 3.07 | 13.13 | **0.048** |
| | Language[a] | | 1.43 | -3.17 | 6.04 | ≈1 |
| | Vision | | 2.83 | -2.22 | 7.88 | ≈1 |
| | Seizure | | 0.52 | -1.45 | 2.50 | ≈1 |
| **Physical score**[b, c] | Motor | 0.34 | 13.79 | 6.58 | 21.00 | **0.012** |
| | Language | | 6.12 | 0.37 | 11.87 | 0.922 |
| | Vision | | -0.44 | -7.77 | 6.88 | ≈1 |
| | Seizure | | -0.25 | -3.35 | 2.84 | ≈1 |
| **Emotional score** | Motor | 0.06 | 2.73 | -3.25 | 8.72 | ≈1 |
| | Language | | -1.62 | -6.72 | 3.49 | ≈1 |
| | Vision | | 6.22 | -0.15 | 12.59 | ≈1 |
| | Seizure[a] | | 0.47 | -2.21 | 3.16 | ≈1 |
| **Social score** | Motor | 0.04 | 3.74 | -3.00 | 10.48 | ≈1 |
| | Language[a] | | -1.77 | -7.86 | 4.32 | ≈1 |
| | Vision | | 4.11 | -2.69 | 10.90 | ≈1 |
| | Seizure | | 1.18 | -1.48 | 3.85 | ≈1 |
| **School score** | Motor | 0.06 | 4.08 | -3.41 | 11.58 | ≈1 |
| | Language | | -5.60 | -11.61 | 0.42 | ≈1 |
| | Vision | | 3.94 | -4.56 | 12.44 | ≈1 |
| | Seizure[a] | | 3.22 | 0.01 | 6.43 | 0.944 |

*P-values are Holm-Bonferroni corrected; obtained from Kenward-Roger adjusted F-tests; considered statistically significant and highlighted in **bold** if less than 0.05.

CI, Confidence intervals are Wald type approximations.

[a]Random slopes for this independent variable included in model.

[b]Random intercepts included in model.

[c]Random slopes for time variable included in model.

drives this relationship. Estimates of the regression parameter in the mixed effects models for the ML score of the Clinical Rating Scale were larger than the 4.5-point MCID for both the PedsQL Total score and Physical domain score [15]. This relationship could suggest that a 1-point change in the ML score provides a MCID in the PedsQL Total score and Physical domain score, highlighting the clinical relevance of the motor and language aspects of CLN2 disease. These findings are unsurprising given that the motor and language domains are thought to most closely reflect disease progression [8, 9].

**Table 8. Results of the multivariate linear mixed effects model comparing the Motor and Language domains of the Clinical Rating Scale to the PedsQL Total score—Study 201/202.**

| Dependent variable | Independent variables | Marginal $R^2$ | Parameter estimate | 95% CI lower limit | 95% CI upper limit | Adjusted p-value* |
|---|---|---|---|---|---|---|
| **PedsQL Total score**[a, b] | Motor | 0.20 | 7.17 | 2.30 | 12.05 | **0.023** |
| | Language | | 2.41 | -1.65 | 6.48 | 0.274 |

*P-values are Holm-Bonferroni corrected; obtained from Kenward-Roger adjusted F-tests; considered statistically significant and highlighted in **bold** if less than 0.05.

CI, Confidence intervals are Wald type approximations.

[a]Random intercepts included in model.

[b]Random slopes for time variable included in model.

By analysing the motor and language domains of the ML score separately, we hoped to understand which of the two domains are most closely correlated to the PedsQL or if they behave in a similar way. Results from the multivariate models suggested that the motor domain alone may be sufficient to explain the PedsQL Total score, which is consistent with the observed relationship between the motor domain and the PedsQL Physical domain. The language, vision and seizure domains of the Clinical Rating Scale, however, did not show a statistically significant association with the PedsQL when analysed individually in linear mixed effects models, which may suggest the PedsQL is not able to capture the full impact of CLN2 disease on patients' QoL. As previously noted, the loss of vision at later stages of the disease is likely to be in part contributing to the lack of a relationship between the vision domain and the PedsQL, as by these later stages, the PedsQL score is already low meaning further changes are likely to be minimal [7, 9]. In addition, there are a number of challenges associated with measuring seizures, for example, the impact of anti-epileptic drugs, that may make identifying a relationship between the seizure domain and the PedsQL more challenging. Furthermore, the seizure domain of the Clinical Rating Scale only captures generalised tonic-clonic (grand mal) seizures; however, it is known that highly variable seizure patterns, consisting of multiple seizure types including myoclonic, tonic, atonic, absence, as well as tonic-clonic seizures, are observed over the course of CLN2 disease [6, 7]. These other seizure types can either directly or indirectly impact QoL of patients; for example, myoclonus is known to contribute to sleep disturbance and reduce QoL for both patients and their families [4, 22]. These results therefore may indicate the need for a disease-specific QoL measure that is better able to capture the highly varied emotional and psychological effects of CLN2 disease and allow a more comprehensive understanding of patients' QoL. Such a disease-specific measure may be better able to capture the QoL impact of, for example, the introduction of a feeding tube, which is often required in more advanced stages of this disease [4].

## Limitations

Analyses presented in this report were not predefined before the start of Study 201/202 and therefore should be considered exploratory only. Furthermore, Study 201/202 was not powered for the comparisons featured in this report and, therefore, results presented cannot provide definitive evidence against hypotheses and should only be used to inform further analysis. As such, these results should be interpreted with care. As with many studies investigating rare diseases, results from statistical significance testing may not have been robust due to the small amount of data available in many of the analyses.

The large number of tests used in this study also meant that a number of standard assumptions associated with regression analyses were not thoroughly tested. Standard assumptions included a linear relationship between variables of interest, independence and homoscedasticity. However, visual inspection of residual plots across all models indicated no violation of these assumptions (results not shown). An additional assumption for all the regression analyses that were performed was that the relationship between the independent and dependent variable does not change over time (study week). When acting as dependent variables, Clinical Rating Scale (Total score) and PedsQL (Total and domain scores), were treated as continuous variables.

For multicollinearity analyses, it was assumed that any multicollinearity between the domains remained constant over time and normality was assumed. It must also be noted that for the multicollinearity analyses, which used data where multiple observations were taken on a single patient, the independent errors assumption was violated. Furthermore, due to there being only one score of zero in the vision domain for Study 201/202, the vision domain was

highly correlated with the intercept in the regression analyses giving uninformative large condition indices. Hence, condition indices for models without the intercept were calculated when the vision domain was included in the analysis.

As previously mentioned, the PedsQL has not been fully validated specifically in this population of patients with CLN2 disease. It should also be noted that the use of the parent-reported proxy version of the PedsQL and the clinician-reported Clinical Rating Scale may result in differing perspectives on patient functioning between the respective completers of each assessment. Please see the S1 File for further information on study limitations.

## Conclusions

This study provides evidence to suggest that each domain of the Clinical Rating Scale provides unique information on disease state and patient functioning. Both the ML score and Total score of the Clinical Rating Scale exhibit a relationship with the PedsQL, a generic QoL measure that has yet to be specifically validated in CLN2 disease. The relationship appears to be driven by the motor domain of the Clinical Rating Scale and the Physical domain of the PedsQL. These findings suggest that motor function is a driver of patients' QoL. The lack of association between the individual language, vision and seizure domains of the Clinical Rating Scale and the PedsQL, suggests that in addition to the PedsQL there may be a need for a more sensitive, disease-specific measure in patients with CLN2 disease to highlight other aspects of QoL not captured by the PedsQL.

## Supporting information

**S1 File. Statistical limitations.**
(DOCX)

## Acknowledgments

The authors thank the patients and their caregivers in addition to the investigators and their teams who contributed to this study. The authors also acknowledge Bryony Langford, MSc and Chris McDonald, PhD, who worked at Costello Medical, UK at the time of the study, for statistical analysis support. Medical writing and editorial assistance in preparing this manuscript for publication was provided by Costello Medical, UK, based on the authors' input and direction. The authors acknowledge Eleanor Thurtle, MChem, Costello Medical, Cambridge, for writing and editorial assistance. PedsQL™, Copyright © 1998 JW Varni, Ph.D. All rights reserved. PedsQL™ contact information and permission to use: Mapi Research Trust, Lyon, France—Internet: https://eprovide.mapi-trust.org and www.pedsql.org.

## Author Contributions

**Conceptualization:** Andrew Olaye, Annabel Griffiths, Thomas Butt.

**Formal analysis:** Tristan Curteis.

**Validation:** Peter Slasor.

**Writing – original draft:** Tristan Curteis, Annabel Griffiths.

**Writing – review & editing:** Nicola Specchio, Paul Gissen, Emily de los Reyes, Andrew Olaye, Charlotte Camp, Annabel Griffiths, Thomas Butt, Jessica Cohen-Pfeffer, Peter Slasor, Zlatko Sisic, Mohit Jain, Angela Schulz.

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
