## [Decision Letter · Decision Letter 0]

2 Feb 2022

PONE-D-21-26733Exploring concurrent validity of the CLN2 Clinical Rating Scale: comparison to PedsQLTM using cerliponase alfa clinical trial dataPLOS ONE

Dear Dr. Nicola Spechio 

Thank you for submitting your manuscript to PLOS ONE. After careful consideration, we feel that it has merit but does not fully meet PLOS ONE’s publication criteria as it currently stands. Therefore, we invite you to submit a revised version of the manuscript that addresses the points raised during the review process.

Please follow the input from the reviewers. 

We look forward to receiving your revised manuscript.

Kind regards,

Rizaldy Taslim Pinzon

Academic Editor

PLOS ONE

https://journals.plos.org/plosone/s/file?id=ba62/PLOSOne_formatting_sample_title_authors_affiliations.pdf"

“BioMarin Pharmaceutical Inc funded this study, and were involved in the design of the study, analysis and interpretation of data, as well as in preparation of this manuscript and the decision to submit this article for publication.

Dr Schulz received grants from the German Federal Ministry of Education and Research (NCL2Treat), European Union's Horizon 2020 Research and Innovation Program (66691), and the European Union Seventh Framework Program (281234) for the Dementia in Childhood consortium which provided data for the historical cohort. Great Ormond Street Hospital and UCL Institute of Child Health Biomedical Research Centre received a grant from the National Institute for Health Research.”

“This study was funded by BioMarin Europe Ltd. The authors thank the patients and their caregivers in addition to the investigators and their teams who contributed to this study. The authors also acknowledge Bryony Langford, MSc and Chris McDonald, PhD, who worked at Costello Medical, UK at the time of the study, for statistical analysis support. Medical writing and editorial assistance in preparing this manuscript for publication was provided by Costello Medical, UK, based on the authors’ input and direction. The authors acknowledge Eleanor Thurtle, MChem, Costello Medical, Cambridge, for writing and editorial assistance. All costs associated with the development of this manuscript were funded by BioMarin Europe Ltd. PedsQLTM, Copyright © 1998 JW Varni, Ph.D. All rights reserved. PedsQLTM contact information and permission to use: Mapi Research Trust, Lyon, France – Internet: https://eprovide.mapi-trust.org and www.pedsql.org.”

“BioMarin Pharmaceutical Inc funded this study, and were involved in the design of the study, analysis and interpretation of data, as well as in preparation of this manuscript and the decision to submit this article for publication.

Dr Schulz received grants from the German Federal Ministry of Education and Research (NCL2Treat), European Union's Horizon 2020 Research and Innovation Program (66691), and the European Union Seventh Framework Program (281234) for the Dementia in Childhood consortium which provided data for the historical cohort. Great Ormond Street Hospital and UCL Institute of Child Health Biomedical Research Centre received a grant from the National Institute for Health Research.”

“PG and NS received no specific funding for this work.

AS received compensation from BioMarin Europe as per clinical trial agreement.

EdlR received research grants from Amicus Therapeutics, BioMarin Pharmaceuticals Inc, and Giving Back Fund.

AO, CC, TB and MJ were employees and shareholders of BioMarin Europe at time of study initiation.

ZS was an employee of BioMarin Europe at time of study initiation.

JCP and PS were employees and shareholders of BioMarin Pharmaceutical Inc at time of study initiation.

TC and AG were employees of Costello Medical at time of study initiation, who provided writing support and editorial assistance which was funded by BioMarin Pharmaceuticals.”

7. PLOS requires an ORCID iD for the corresponding author in Editorial Manager on papers submitted after December 6th, 2016. Please ensure that you have an ORCID iD and that it is validated in Editorial Manager. To do this, go to ‘Update my Information’ (in the upper left-hand corner of the main menu), and click on the Fetch/Validate link next to the ORCID field. This will take you to the ORCID site and allow you to create a new iD or authenticate a pre-existing iD in Editorial Manager. Please see the following video for instructions on linking an ORCID iD to your Editorial Manager account: https://www.youtube.com/watch?v=_xcclfuvtxQ.

Additional Editor Comments:

This is an interesting topic. Many methodological flaws in your study. Please follow the recommendations and input from the reviewer.

Reviewers' comments:

Reviewer's Responses to Questions

**Comments to the Author**

1. Is the manuscript technically sound, and do the data support the conclusions?

Reviewer #1: No

2. Has the statistical analysis been performed appropriately and rigorously? 

Reviewer #1: No

3. Have the authors made all data underlying the findings in their manuscript fully available?

Reviewer #1: No

4. Is the manuscript presented in an intelligible fashion and written in standard English?

Reviewer #1: Yes

5. Review Comments to the Author

Reviewer #1: PONE-D-21-26733: statistical review.

SUMMARY. This is a study of a clinical rating scale that evaluates progression of an ultra-rare neurodegenerative disorder (the CLN2 disease) with late infantile onset. It investigates the cross-correlations between the four domains of the scale (motor, language, vision and seizure) and the association between this scale and the domains of the Pediatric Quality of Life PedsQL score. The statistical analysis relies on marginal correlations between the domain-specific scores of the clinincal scale under study and on the estimation of linear mixed models, where PedsQL scores are predicted by clinical scale scores. I have major concerns about the methods employed (see the major issues below), which require a complete revision of the statistical analysis. Appended are further specific points that should be considered by the authors.

MAJOR ISSUES

1. Cross-correlations between domain-specific scores are computed by a battery of Spearman correlation indexes. This method is appropriate because the scores are ordinal variables. However, p-values (or standard errors) are not provided, complicating the interpretation of Table 2. P-values must be provided. Due to the small sample size, I wouldn't rely on the asymptotic distributions of the Spearman rho, though. I'd recommend a bootstrap approach to compute p-values.

2. I guess that the computation of the VIF and the Condition index were based on the Pearson correlations (although this was not specified, please clarify). These measures are typical of collinearity analysis in multivariate continuous data, but they are certainly not appropriate here, where the data are multivariate ordinal. This part of the analysis is hence incosistent with the Spearman analysis and should be removed.

3. The association between the CLN2 scale and the domains of the Pediatric Quality of Life PedsQL score is based on the assumption of normally distributed PedsQL scores. The authors should provide evidence that these scores are normally distributed. Fitting a normal linear model to non-normal data could lead to biased p-values.

4. Page 8 "Clinical Rating Scale data were collected every 8 weeks and PedsQL data every 12 weeks, leading to

16 concurrent data for both measures being available every 24 weeks". Please carify whether the analysis (correlations and linear mixed models) has been performed by either using only the concurrent data and discarding the nonoccurrent observations (an unnecessary waste of information...) or including all the data and treating the observations as missing at random (a hypothesis that must be motivated).

5. Page 9 "a selection of random effect structures was considered, including only random intercepts, only random slopes, and both random intercepts and slopes." Two points to be clarified, here. First, the criterion used to select the random effect stucture should be provided. Second, tables 6-8 should indicate which random structure has been selected for each model, to allow replicability of the results (an important requirement of the Journal).

SPECIFIC POINTS

1. As remarked in major issue no 1, Spearman correlation is an appropriate method in the paper setting. However, it provides very little information about associations between scores, becuase it does not account for the longitudinal structure of the data. The state-of-the art approach would be a log-linear model that investigates the interations between multiple scores. Log-linear models can be easily estimated by R, which is the software used by the authors.

2. Introduction, line 14. What do the authors mean by "median life expectancy"? Life expectancy is already a mean. Hence I don't understand what is the median of a mean. Please carify how this summary has been computed.

3. It seems that many references in the text went lost during the final compilation of the manuscript ...

4. Although the authors have declared that the data are available without restrictions, data are not provided. Data should be attached as supplementary information, in a publicly available format (txt or xls). This is also an important requirement of the Journal.

6. PLOS authors have the option to publish the peer review history of their article (what does this mean?). If published, this will include your full peer review and any attached files.

Reviewer #1: No

---

## [Author Response · Author response to Decision Letter 0]

8 Sep 2022

Dr Nicola Specchio 

Department of Neuroscience 

Bambino Gesù Children’s Hospital, IRCCS, 

Piazza di Sant'Onofrio, 4

Rome, 00165 

Italy

Email address: nicola.specchio@gmail.com

15 June 2022

Handling Editor: Rizaldy Taslim Pinzon

PLOS ONE

RE: Exploring concurrent validity of the CLN2 Clinical Rating Scale: comparison to PedsQLTM using cerliponase alfa clinical trial data (Manuscript ID: PONE-D-21-26733)

Dear Dr Pinzon,

Many thanks for providing us with the peer reviewers’ feedback on the above manuscript. We have now revised the manuscript to implement the comments and have included a detailed point-by-point response to each comment within this letter (see below). We have also prepared the revised manuscript using tracked changes to aid review.

We hope that the individual responses below, and revisions to the manuscript, address the concerns raised by the peer reviewers, and that the manuscript is now suitable for publication in PLOS ONE.

We look forward to your reply.

Yours sincerely,

Dr Nicola Specchio, on behalf of all co-authors

 

Major issues

Cross-correlations between domain-specific scores are computed by a battery of Spearman correlation indexes. This method is appropriate because the scores are ordinal variables. However, p-values (or standard errors) are not provided, complicating the interpretation of Table 2. P-values must be provided. Due to the small sample size, I wouldn't rely on the asymptotic distributions of the Spearman rho, though. I'd recommend a bootstrap approach to compute p-values.

Response: Thank you for your suggestion. The p-values have now been generated to support the results in Table 2 via bootstrapping 1,000 runs as requested.

I guess that the computation of the VIF and the Condition index were based on the Pearson correlations (although this was not specified, please clarify). These measures are typical of collinearity analysis in multivariate continuous data, but they are certainly not appropriate here, where the data are multivariate ordinal. This part of the analysis is hence inconsistent with the Spearman analysis and should be removed.

Response: Thank you for your feedback. We can confirm that computation of the VIF and the Condition index was based on the Pearson correlations. While we agree that strictly speaking, quality of life measures are indeed ordinal, these tend to be treated as continuous in regression modelling in most literature. Moreover, our two outcomes, CLN2QL and PedsQL total score, spread across a wide range of values between 0 and 100 (i.e. not heavily skewed) hence we believe that treating these as continuous in this context is viable.

The association between the CLN2 scale and the domains of the Pediatric Quality of Life PedsQL score is based on the assumption of normally distributed PedsQL scores. The authors should provide evidence that these scores are normally distributed. Fitting a normal linear model to non-normal data could lead to biased p-values.

Response: Thank you for your feedback. The normality assumption of linear regression and linear mixed-effects models generally applies to the residuals rather than the dependent variables. We have thus assessed this using diagnostic plots of within-group residuals versus fitted values in the course of our analyses, which by inspection indicate that the assumption of a constant error variance with an approximate normal distribution holds for all analyses, and that it is appropriate to fit linear mixed effects models to these data. We have not included these plots in the manuscript given the large number of plots generated, however we have included the following clarification in the manuscript on page 20, line 30: “However, visual inspection of residual plots across all models indicated no violation of these assumptions (results not shown).”

Page 8 "Clinical Rating Scale data were collected every 8 weeks and PedsQL data every 12 weeks, leading to 16 concurrent data for both measures being available every 24 weeks". Please clarify whether the analysis (correlations and linear mixed models) has been performed by either using only the concurrent data and discarding the nonconcurrent observations (an unnecessary waste of information...) or including all the data and treating the observations as missing at random (a hypothesis that must be motivated).

Response: Thank you for your suggestion. We can confirm that concurrent data was used here for both correlations and the models. In terms of dealing with missingness, observations have been treated as missing at random. The rationale behind this approach is based on data collection in the trial, where outcomes were reported via proxy (e.g. parent responding on the child’s behalf). Subsequently, likelihood of receiving a response from a parent (as opposed to the child in question) is much more likely to be independent from good/poor health of the child. We evaluated this assumption by comparing clinical rating scale scores immediately before a missing entry with overall clinical rating scores and found no signs of association, suggesting that this assumption is valid. 

Page 9 "a selection of random effect structures was considered, including only random intercepts, only random slopes, and both random intercepts and slopes." Two points to be clarified, here. First, the criterion used to select the random effect structure should be provided. Second, tables 6-8 should indicate which random structure has been selected for each model, to allow replicability of the results (an important requirement of the Journal).

Response: Thank for your suggestion. We can confirm that the Akaike information criterion was used to select the final model and have highlighted this in the manuscript. The model structures for each of the models have been identified for each analysis, and we have updated Table 6, Table 7 and Table 8 in the manuscript to include these. We have described for each model whether a random intercept, random slope for the time variable, random slope for the dependent variable, or a combination of these were used.

Specific Points

As remarked in major issue no 1, Spearman correlation is an appropriate method in the paper setting. However, it provides very little information about associations between scores, because it does not account for the longitudinal structure of the data. The state-of-the art approach would be a log-linear model that investigates the interactions between multiple scores. Log-linear models can be easily estimated by R, which is the software used by the authors.

Response: Thank you for your suggestion. As log-linear models are commonly used for count data, rather than continuous/ordinal data, this would not yield statistics that relay information on the relationships between each domain in this case. Subsequently, we felt that this would not be appropriate for the data presented in the manuscript. We acknowledge that ideally, the longitudinal structure of the data would be accounted for when producing the correlations, and that this is a limitation of this preliminary analysis. However, we consider that the correlations, with the update to include the p-values, adequately indicate which domains capture similar information in the first instance, and in subsequent analyses, the longitudinal structure of the data is accounted for by linear mixed effect models. 

Introduction, line 14. What do the authors mean by "median life expectancy"? Life expectancy is already a mean. Hence I don't understand what is the median of a mean. Please clarify how this summary has been computed.

Response: Many thanks for flagging this. We have now updated the text in the Introduction to align with Nickel et al 2018, which now reads “Ultimately, patients with CLN2 disease experience reduced life expectancy, with a median age at death reported as 10.0 years”.

It seems that many references in the text went lost during the final compilation of the manuscript ...

Response: Many apologies for this oversight here and thank you for flagging. Unfortunately, during the upload to the manuscript submission site, some reference links were broken. We have ensured that all links are fixed during resubmission of the manuscript.

Although the authors have declared that the data are available without restrictions, data are not provided. Data should be attached as supplementary information, in a publicly available format (txt or xls). This is also an important requirement of the Journal.

Response: The de-identified individual participant data that underlie the results reported in this article (including text, tables, figures, and appendices) will be made available together with the research protocol and data dictionaries, for non-commercial, academic purposes. Additional supporting documents may be available upon request. 

Investigators will be able to request access to these data and supporting documents via a data sharing portal beginning 6 months and ending 2 years after publication. Data associated with any ongoing development program will be made available within 6 months after approval of relevant product. Requests must include a research proposal clarifying how the data will be used, including proposed analysis methodology. 

Research proposals will be evaluated relative to publicly available criteria available at www.biomarin.com/patients/publication-data-request/ to determine whether access will be given, contingent upon execution of a data access agreement with BioMarin Pharmaceutical Inc.

Funding statement: Please could the Funding Statement be updated as follows, “BioMarin Pharmaceutical Inc funded this study, and were involved in the design of the study, analysis and interpretation of data, as well as in preparation of this manuscript and the decision to submit this article for publication. Medical writing and editorial assistance in preparing this manuscript for publication was provided by Costello Medical, UK, and funded by BioMarin Pharmaceutical Inc.” Please note that all funding information has been removed from the Acknowledgements section of the manuscript.

Competing interests statement: Please could the Competing Interests Statement be updated as follows, “PG, NS: No competing interests to declare; AS: Received compensation from BioMarin Europe as per clinical trial agreement; EdlR: Research grants from Amicus Therapeutics, BioMarin Pharmaceutical Inc., Giving Back Fund; Consultancy: BioMarin Pharmaceuticals Inc.; AO, CC, TB, and MJ: Employees and shareholders of BioMarin Europe; ZS: Employee of BioMarin Europe; TC and AG: Employees of Costello Medical; JCP and PS: Employees and shareholders of BioMarin Pharmaceutical Inc. This does not alter our adherence to PLOS ONE policies on sharing data and materials.”

---

## [Editor Report · Decision Letter 1]

21 Sep 2022

PONE-D-21-26733R1Exploring concurrent validity of the CLN2 Clinical Rating Scale: comparison to PedsQLTM using cerliponase alfa clinical trial dataPLOS ONE

Dear Dr. Specchio

Thank you for submitting your manuscript to PLOS ONE. After careful consideration, we feel that it has merit but does not fully meet PLOS ONE’s publication criteria as it currently stands. Therefore, we invite you to submit a revised version of the manuscript that addresses the points raised during the review process.  Please improve the quality of the figure. Please make the limitations and conclusion of your study more concise and straightforward.  Please submit your revised manuscript by  Nov 05 2022 11:59PM.  If you will need more time than this to complete your revisions, please reply to this message or contact the journal office at plosone@plos.org. Please include the following items when submitting your revised manuscript:A rebuttal letter that responds to each point raised by the academic editor and reviewer(s). You should upload this letter as a separate file labeled 'Response to Reviewers'.A marked-up copy of your manuscript that highlights changes made to the original version. You should upload this as a separate file labeled 'Revised Manuscript with Track Changes'.An unmarked version of your revised paper without tracked changes. You should upload this as a separate file labeled 'Manuscript'.

We look forward to receiving your revised manuscript.

Kind regards,

Rizaldy Taslim Pinzon

Academic Editor

PLOS ONE

Additional Editor Comments:

Thank you for your prompt reply and revisions. Please improve the quality of the figure. Please make the limitations and conclusion of your study more concise and straightforward.
---

## [Author Response · Author response to Decision Letter 1]

28 Nov 2023

Please improve the quality of the figure. 

Response: We have now redrawn the figure as requested.

Please make the limitations and conclusion of your study more concise and straightforward. 

Response: We have updated the limitations and conclusion section to make this more concise. Please note, the detailed information regarding the statistical limitations has been moved to a supplementary materials file, should readers still wish to see this information.

---

## [Decision Letter · Decision Letter 2]

2 Apr 2024

Exploring concurrent validity of the CLN2 Clinical Rating Scale: comparison to PedsQLTM using cerliponase alfa clinical trial data

PONE-D-21-26733R2

Dear Dr. Specchio,

We’re pleased to inform you that your manuscript has been judged scientifically suitable for publication and will be formally accepted for publication once it meets all outstanding technical requirements.

Kind regards,

Markus Ries, MD PhD MHSc MA FCP

Academic Editor

PLOS ONE

Additional Editor Comments (optional):

Please note that some reference links in the text show error codes.

Reviewers' comments:

Reviewer's Responses to Questions

**Comments to the Author**

1. If the authors have adequately addressed your comments raised in a previous round of review and you feel that this manuscript is now acceptable for publication, you may indicate that here to bypass the “Comments to the Author” section, enter your conflict of interest statement in the “Confidential to Editor” section, and submit your "Accept" recommendation.

Reviewer #1: All comments have been addressed

2. Is the manuscript technically sound, and do the data support the conclusions?

Reviewer #1: (No Response)

3. Has the statistical analysis been performed appropriately and rigorously? 

Reviewer #1: (No Response)

4. Have the authors made all data underlying the findings in their manuscript fully available?

Reviewer #1: (No Response)

5. Is the manuscript presented in an intelligible fashion and written in standard English?

Reviewer #1: (No Response)

6. Review Comments to the Author

Reviewer #1: (No Response)

7. PLOS authors have the option to publish the peer review history of their article (what does this mean?). If published, this will include your full peer review and any attached files.

Reviewer #1: No

---

## [Editor Report · Acceptance letter]

10 May 2024

PONE-D-21-26733R2 

PLOS ONE

Dear Dr. Specchio, 

I'm pleased to inform you that your manuscript has been deemed suitable for publication in PLOS ONE. Congratulations! Your manuscript is now being handed over to our production team.

Kind regards, 

on behalf of

Professor Markus Ries 

Academic Editor

PLOS ONE